# Commercially Available *Viola odorata* Oil, Chemical Variability and Antimicrobial Activity

**DOI:** 10.3390/molecules28041676

**Published:** 2023-02-09

**Authors:** Ané Orchard, Tasneem Moosa, Nabeelah Motala, Guy Kamatou, Alvaro Viljoen, Sandy van Vuuren

**Affiliations:** 1Department of Pharmacy and Pharmacology, School of Therapeutic Sciences, Faculty of Health Sciences, University of the Witwatersrand, Johannesburg 2193, South Africa; 2Department of Pharmaceutical Sciences, Faculty of Science, Tshwane University of Technology, Private Bag X680, Pretoria 0001, South Africa; 3SAMRC Herbal Drugs Research Unit, Department of Pharmaceutical Sciences, Private Bag X680, Pretoria 0001, South Africa

**Keywords:** minimum inhibitory concentration, chemical variability, fractional inhibitory concentration, combinations, antimicrobial, gas chromatography–mass spectrometry, *Viola odorata*

## Abstract

*Viola odorata* L. oil is frequently recommended in the aromatherapeutic literature for treating respiratory, urinary, and skin infections; however, antimicrobial evidence is lacking. In addition, in aromatherapy, combinations of essential oils are predominantly utilized with the goal of achieving therapeutic synergy, yet no studies investigating the interaction of essential oil combinations with *V. odorata* oil exists. This study thus aimed to address these gaps by investigating the antimicrobial activity of three *Viola odorata* oil samples, sourced from different suppliers, independently and in combination with 20 different commercial essential oils, against micro-organisms involved in respiratory, skin, and urinary tract infections associated with global resistance trends. These pathogens include several of the ESKAPE pathogens (*Enterococcus faecium, Staphylococcus aureus, Klebsiella pneumoniae, Acinetobacter baumannii, Pseudomonas aeruginosa* and *Enterobacter* spp.) The chemical profile of the oils was determined using gas chromatography coupled with mass spectrometry. The minimum inhibitory concentrations (MIC) were determined using the broth micro-dilution method. The interactive profiles for the combinations were assessed by calculating the fractional inhibitory concentration index (ΣFIC). The main compounds varied across the three samples, and included phenethyl alcohol, isopropyl myristate, 2-nonynoic acid, methyl ester, α-terpineol, α-cetone, and benzyl acetate. The *V. odorata* oil samples displayed overall poor antimicrobial activity when tested alone; however, the antimicrobial activity of the combinations resulted in 55 synergistic interactions where the combination with *Santalum austrocaledonicum* resulted in the lowest MIC values as low as 0.13 mg/mL. The frequency of the synergistic interactions predominantly occurred against *Klebsiella pneumoniae*, *Pseudomonas aeruginosa*, *Acinetobacter baumannii*, and *Enterococcus faecium* with noteworthy MIC values ranging from 0.25–1.00 mg/mL. This study also reports on the variability of *V. odorata* oils sold commercially. While this warrants caution, the antimicrobial benefit in combination provides an impetus for further studies to investigate the therapeutic potential.

## 1. Introduction

The use of natural products is of particular interest to “green consumers” who have a preference for eco-friendly products which are naturally derived [1]. Essential oils have gained popularity over the years, originating from the practice of aromatherapy, which is an alternative medical therapy incorporating the use of volatile/plant essential oils for disease management [2,3,4,5]. Essential oils have further been shown to demonstrate antimicrobial activity, including against antimicrobial-resistant strains [6,7,8], and they have been proven to display activity in clinical studies [9,10].

*Viola odorata* L., commonly known as English or sweet violet, is a member of the Violaceae family and is indigenous to Europe, North Africa, and western Asia [11]. The therapeutic potential of the plant has gained attention in recent years [12]. *Viola odorata* essential oil is recommended for the treatment of various ailments and is also frequently recommended in combination with other commercial essential oils (Table 1). Combining essential oils is a common practice as it is believed that the different properties contributed by each essential oil may achieve a synergistic therapeutic benefit [13]. Despite the use of this oil in aromatherapy, scientific evidence validating the antimicrobial activity alone and in combination against the pathogens involved in the recommended infectious diseases to be treated by this oil (Table 1) is deficient, which is surprising if one considers how often it is cited and recommended for use in combination [12,14,15,16,17]. Added to that, *V. odorata* is also often available and used as an absolute oil which is a highly aromatic liquid extracted via solvent extraction as opposed to distillation with the perceived benefit of a stronger scent. This may result in chemical variability between the absolute and essential oil [18].

While there have been some studies on the antimicrobial activity and variation in the chemistry of *V. odorata* [19,20], the majority of the studies were acutely focused on simple analysis (disk diffusion assays) [19,21]. Further studies that depicted antimicrobial activity were limited to other *Viola* spp., such as *V. calcarata* and *V. dubyana* [22] or the extracts (non-volatiles) of the *V. odorata* plant [23,24,25,26,27,28].

In terms of chemistry, the variability of major compounds of *V. odorata* oil appears to vary across previous studies. Pentane 2,3,4-trimethyl, N-hexadecanoic acid, 10-undecyn-1-l and pentadecanoic acid is reported as the main compounds in the oil by one study [20], and butyl-2-ethylhexylphtalate and 5,6,7,7a-tetrahydro-4,4,7a-trimethyl-2(4H)-benzofuranone was reported in another [19]. This indicates that investigations of this natural product may be beneficial by including multiple collections to identify how variable the chemistry is, and its effects on the antimicrobial activity. Based on the variation of previously reported oil samples, emphasis is placed on the importance of analytical (GC-MS) data to accompany the bio-activity results. Thus, this study aimed to investigate the chemical composition and the antimicrobial profile of three different commercially available *Viola odorata* oil samples, independently and in combination with a range of commercial essential oils.

## 2. Results and Discussion

### 2.1. Chemical Analysis

The chemical analysis of three *V. odorata* oil samples (designated 1–3) is shown in Table 2. The chemical analysis of the commercials oils that were combined with the *V. odorata* oil samples has been previously reported [6,29].

While all the *V. odorata* oil samples were commercially available, a clear quantitative and qualitative variation exists across all three oils. The *V. odorata* 1 and *V. odorata* 2, procured from The United States, comprises of a high percentage of the compound phenethyl alcohol, which is a known essential oil compound with a distinct floral note, found in various essential oils such as *Rosa damascena* Mill. (rose otto) [30,31]. Oil sample 1 also had 2-Nonynoic acid, methyl ester (also known as methyl 2-nonynoate [32]) as a main compound, which is responsible for the floral scent of *V. odorata* [33]. Oil sample 2 had isopropyl myristate as a main compound, which suggests possible adulteration. Isopropyl myristate is a fatty acid derivative composed of isopropyl alcohol and myristic acid (fatty acid), which is a common additive in cosmetics [34]. This is not the first study to identify variations in the chemistry in *Viola* spp., as a study on the chemical composition of *V. calcarata* and *V. dubyana* identified fatty acids in the oil samples which had been distilled by the authors of that study [22]. The main compounds of our oil sample 3 were α-terpineol, a known essential oil compound, benzyl acetate which is a natural product found in *Vitis rotundifolia* Michx. and *Tanacetum parthenium* L., and lastly α-cetone [35]. Other studies have reported dominant compounds such as 1-hexadecene, 1-octadecene, 1-ecosene and hexadecanoic acid in *V. odorata* from Toulouse, France [36], butyl-2-ethylhexylphtalate, 5,6,7,7 α-tetrahydro-4,4,7 α-trimethyl-2(4 H)-benzofuranone from Iran [19], pentane 2,3,4-trimethyl, N-hexadecanoic acid, 10-undecyn-1-l, pentadecanoic acid from Kermanshah [20,37], 1-phenyl butanone, linalool, benzyl alcohol, α-cadinol, globulol, viridiflorol from Tunisia [38], and terpineol, benzyl acetate, methyl salicylate, eugenol, pentadecanoic acid ethyl ester, pentaoxahexadecan-1-ol, tetraoxahexadecan-1-ol, octadecadienal, octadecatrienoic acid ethyl ester, pentaoxanonadecan-1-ol and hexadecanoic acid, all from Egypt [39]. These previous studies have all identified different major compounds across the *V. odorata* oil samples, and this trend continues with the findings reported in this study. However, several minor compounds appear to be present here or across some of the reported studies such as α-terpineol, methyl benzoate, geraniol, terpinen-4-ol, α-pinene, β-ionone, α-hexyl cinnamaldehyde [19,38,40]. Unlike most commercial essential oils, this variation across samples makes it difficult to define what can be expected from the chemistry of *V. odorata*. It is known that the chemical composition may vary according to harvest and geographical location of the essential oils; however, this rampant variation across *V. odorata* oil would be in the percentage of compounds composition, as opposed to variations in types of compounds present.

What was also not common for these tested *V. odorata* oil samples is the presence of a synthetic compound such as Lilial present in two of the samples that are sold as pure essential oils. This would indicate adulteration of the types of products available to consumers where synthetic compounds are added to the oils to enhance certain properties.

### 2.2. Antimicrobial Activity

The minimum inhibitory concentration (MIC) activity of the three *V. odorata* oil samples and the combination with commercial essential oils is given in Table 3. Noteworthy antimicrobial activity was observed for *V. odorata* 1 and *V. odorata* 2 (USA samples) against *Cutibacterium acnes* ATCC 6919 and *Candida albicans* ATCC 10231 (MIC values range 0.50–0.67 mg/mL). *Viola odorata* 3 (derived from Turkey) displayed noteworthy antimicrobial activity against *Staphylococcus epidermidis* ATCC 14990. The overall antimicrobial activity, however, was moderate to poor, which is surprising considering the popularity of this oil (Table 1). What is interesting to observe, however, is the correlation between the geographical location (1 and 2) and consistent antimicrobial activity pattern.

Several of the main compounds across the chemotypes, such as phenethyl alcohol, α-terpineol and terpinen-4-ol have been reported to display bactericidal activity against micro-organisms such as *Staphylococcus aureus*, *Enterococcus faecium*, *Escherichia coli* and *Pseudomonas aeruginosa* [41,42,43]. Previous studies investigating the *Viola* spp. essential oils were limited to investigating the antimicrobial activity (minimum inhibitory concentrations) against *Klebsiella pneumoniae* (0.13 mg/mL), *Staphylococcus epidermidis* (0.50 mg/mL) and *Bacillus subtilis* (0.50–31.00 mg/mL) [19,20]. Against *K. pneumoniae* and *S. epidermidis*, *V. odorata* was previously reported to display stronger antimicrobial activity than that reported in this study. The difference in antimicrobial activity of *V. odorata* in the current study and that reported in the literature is not surprising, considering the difference in chemistry between this study and those previously reported.

For the other commercial oils, *Cutibacterium acnes* ATCC 6919 was the most susceptible micro-organism, with 16 essential oils inhibiting the reference strain at noteworthy concentrations (0.09–1.00 mg/mL). *Klebsiella pneumoniae* ATCC 13883 was clearly the most resilient of the strains tested. The *Santalum* spp. and *Vetiveria zizanioides* are highlighted as essential oils with predominantly noteworthy antimicrobial activity across the selected micro-organisms. This is congruent with previous findings [6].

While *V. odorata* mostly displayed poor antimicrobial activity when tested independently, in combination (the preferred application of essential oil use in aromatherapy), there were several synergistic interactions where the antimicrobial activity was enhanced (Table 4). A summary of the synergy is provided in Figure 1, where *V. odorata* samples 1 and 2, which both have similar chemical profiles, have 14 synergistic interactions, while the third sample having a different chemical profile, has almost double the synergistic interactions and least number of antagonistic interactions. Figure 1 provides a summary of the combinations with each of the *V. odorata* samples analysed independently and then grouped to include all three *V. odorata* samples.

The frequency of the synergistic interactions is mostly against the reference organisms *K. pneumoniae*, *P. aeruginosa*, *A. baumannii*, and *E. faecium*. Each of these form part of the resistant ESKAPE pathogens [44] and thus hold the potential to combat antimicrobial resistance. The former three are also Gram-negative micro-organisms, which have been reported to generally be less susceptible to the inhibition of essential oils [6].

The essential oils *L. cubeba*, *L. scoparium*, *S. album*, and *S. austrocaledonicum* were each involved in at least five synergistic combinations with the *V. odorata* samples. A previous study, albeit different essential oils, has also noted the frequency of *S. austrocaledonicum* in synergistic interactions [45].

## 3. Materials and Methods

### 3.1. Essential Oil Selection

Three commercially available *V. odorata* oil samples (*V. odorata* 1, *V. odorata* 2 and *V. odorata* 3) samples were investigated. The selection of the 20 commercial essential oils investigated in combination with the *V. odorata* samples was based on the aroma-therapeutic literature available to the layman [14,15,16,17], as well as commercially popular essential oils. A selection of commercial oils based on a range of antimicrobial activity, representative of noteworthy, moderate, and poor antimicrobial activity, was selected from our previous studies [6,29]. The essential oils were obtained from Scatters Oils (Johannesburg, South Africa) and Prana Monde (Midrand, South Africa). 

### 3.2. Chemical Analysis

Gas chromatography coupled with mass spectrometry (GC-MS) was used to analyse the chemistry of the three *V. odorata* oil samples. The GC system (Agilent 6890 N GC, Santa Clara, CA, USA) was coupled directly to a 5973 MS equipped with an HP-Innowax polyethylene glycol column (60 m × 250 μm i.d. × 0.25 μm film thickness). A volume of 1 μL was injected (using a split ratio of 200:1 in hexane) with an auto-sampler at 24.79 psi and an inlet temperature of 250 °C. The GC oven temperature was maintained at a temperature of 60 °C for 10 min, then 220 °C at a rate of 4 °C/min for 10 min, followed by a temperature of 240 °C at a rate of 1 °C/min. Helium, the carrier gas, was at a constant flow of 1.2 mL/min. Spectra were obtained on electron impact at 70 eV, scanning from m/z 35 to 550. The percentage composition of the individual components was then quantified by integration measurements using flame ionization detection (FID, 250 °C), and n-alkanes were used as reference points in the calculation of relative retention indices (RRI). Component identifications were made by comparing mass spectra from the total ion chromatogram and retention indices using NIST^®^ Version 2.2 and Mass Finder^®^ Version 4 libraries. The chemical analysis of the 20 commercials oils that were combined with the *V. odorata* samples was previously reported [6,29].

### 3.3. Antimicrobial Analysis

#### 3.3.1. Preparation of Cultures

The ten micro-organisms tested were from the American Type Culture Collection (ATCC) available within the Department of Pharmacy and Pharmacology of the University of the Witwatersrand (Johannesburg, South Africa). These included reference strain pathogens related to the infections for which *V. odorata* oil is most commonly recommended, such as respiratory infections (*Klebsiella pneumoniae* ATCC 13883, *Streptococcus pyogenes* ATCC 12344), skin infections (*Pseudomonas aeruginosa* ATCC 27858, *Escherichia coli* ATCC 25922, *Staphylococcus aureus* ATCC 25924, *Staphylococcus epidermidis* ATCC 14990 and *Cutibacterium acnes* ATCC 6919), opportunistic nosocomial infections (*Acinetobacter baumannii* ATCC 17606 and *Enterococcus faecium* ATCC 8739) and a fungal pathogen reference strain (*Candida albicans* ATCC 10231). The aerobic bacteria and *C. albicans* were cultured in Tryptone Soya broth (TSB) (Oxoid, Basingstoke, UK) at 37 °C for 24 h and 48 h, respectively. *Cutibacterium acnes* was inoculated into Thioglycolate broth (TGB) (Oxoid) under anaerobic conditions using a CO_2_ incubator (8.4% CO2) for seven days at 37°C, and *S. pyogenes* was inoculated into a Haemophilus broth (HB) (Oxoid), supplemented with nicotinamide adenine dinucleotide (NAD) (Oxoid), at 37 °C for 24 h. A waiver for the use of these micro-organisms was granted by The University of the Witwatersrand Human Research Ethics Committee (Reference W-CJ-131026-3).

#### 3.3.2. The Minimum Inhibitory Concentration (MIC)

The broth microdilution method was used to determine the MIC of the essential oils alone and in combination [29]. First, 100 µL of appropriate media was aseptically added into all 96 wells of a micro-titre plate, followed by 100 µL of the sample (or 50 µL *V. odorata* samples with 50 µL of a commercial essential oil for the combinations), diluted to a starting concentration of 32.00 mg/mL in acetone, and added into the first well. This was serially diluted descending down the micro-titre plate. Antimicrobial susceptibility was confirmed using 0.01 mg/mL ciprofloxacin (Sigma Aldrich^®^, St. Louis, MO, USA) (for bacteria) or 0.10 mg/mL amphotericin B (Sigma Aldrich^®^) (for *C. albicans*). A negative control of 32.00 mg/mL water in acetone was included to ensure the antimicrobial activity was not due to the solvent. The respective growth media with the reference strain was also included to ensure microbial viability. After the preparation of an approximate inoculum concentration of 1 × 10^6^ colony forming units per ml (CFU/mL) for each micro-organism, 100 µL was added to each well. The sterile adhesive sealing film was used to seal each micro-titre plate, and the plates were incubated accordingly. After incubation, 40 µL of 0.04% (*w/v*) *p*-iodonitrotetrazolium violet solution (INT) (Sigma Aldrich^®^) was added to each well. Microbial growth was indicated by a colour change of the indicator to pink or purple, and the lowest concentration displaying no growth was taken as the MIC. Each sample was tested in triplicate, and the average was taken as the MIC value. An MIC value less than or equal to 1.00 mg/mL is considered noteworthy [13]. The fractional inhibitory concentration index (ΣFIC) was calculated for the combinations according to Equation (1) [46];
(1)$$\mathrm{FIC}\;(i)=\frac{\;\left(a\text{*}\right)\mathrm{combined}\;\mathrm{with}\;\left(b\text{*}\right)}{\;\left(a\right)\;\mathrm{independently}\;}\;\mathrm{FIC}\;(\mathrm{ii})=\frac{\;\left(b\right)\;\mathrm{combined}\;\mathrm{with}\;\left(a\right)}{\;\left(b\right)\;\mathrm{independently}\;}\;\Sigma \mathrm{FIC}\;=\;\mathrm{FIC}\;(i)\;+\;\mathrm{FIC}\;(\mathrm{ii})$$
* where (a) is the MIC value of the *V. odorata* oil sample in the combination and (b) is the MIC value of the commercial essential oil.

An ΣFIC of ≤0.5 was interpreted as synergy, an ΣFIC of between >0.5 and ≤1.0 was indicative of an additive interaction, >1.0–≤4.0 indicated indifference and >4.0 indicated antagonism [46].

## 4. Conclusions

While *V. odorata* oil is often cited in therapeutic aromatherapy to be used in combination, this study (to the best of our knowledge) is the first to explore the antimicrobial effects in combination. *Santalum austrocaledonicum* combined with the *V. odorata* oil samples could be identified as the combinations that resulted in predominantly noteworthy antimicrobial activity, having the highest frequency of synergistic interactions. The frequency of synergy with low MIC values highlights the potential of these oils and their combinations for further research against antimicrobial-resistant strains.

This study is also the first to highlight variations between the *V. odorata* oil samples sold commercially and emphasizes the importance of including the GC-MS data within aromatic product studies. While some compounds identified appear to be of unnatural origin, it is important to note that these are sold and used for aromatherapy, and whether they are true essential oils or not, the therapeutic value is of importance if used for medical practises. Future studies investigating the comparative chemical profiles of *V. odorata* oils across multiple sources are recommended to allow for a better understanding of what is considered acceptable and non-toxic.

## Figures and Tables

**Figure 1 molecules-28-01676-f001:**
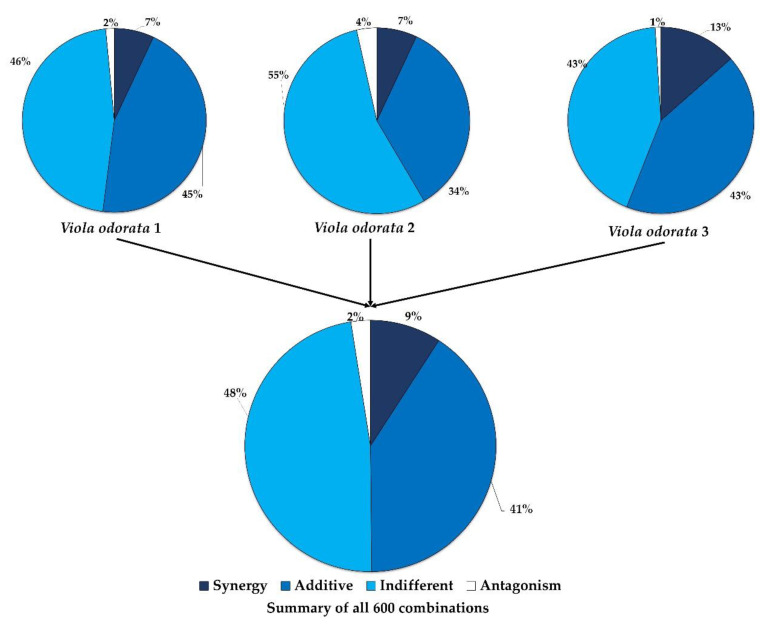
Summary of interactive profile of combinations.

**Table 1 molecules-28-01676-t001:** Use of *V. odorata* oil in the aromatherapeutic literature and recommended combinations [12,14,15,16,17].

System	Indicated Use	Recommended Combinations
Respiratory	Bronchitis, chesty and painful coughs, sore throat and throat infections, inflammation of throat and pleurisy, irritating coughs, whooping cough.	*Boswellia sacra, Cedrus deodara, Cinnamomum camphora, Citrus bergamia, Commiphora myrrha, Cymbopogon winterianus, Cymbopogon nardus, Dalbergia maritima, Eucalyptus globulus, Helichrysum italicum, Hyssopus officinalis, Jasminum polyanthum, Lavandula angustifolia, Melaleuca alternifolia, Myrtus communis, Rosa damascena, Rosmarinus officinalis, Salvia triloba, Santalum album, Santalum austrocaledonicum, Syncarpia glomlifera, Thymus vulgaris, Verbena officinalis, Zingiber officinale.*
Urinary	Purging effect on urine, cystitis.	*Boswellia sacra, Cymbopogon nardus, Dalbergia maritima, Jasminum polyanthum, Lavandula angustifolia, Myrtus communis, Rosa damascena, Santalum album, Verbena officinalis.*
Skin and associated conditions	Acne, bruises, eczema, inflammation, infections, ulcers, and wounds.	*Achillea millefolium, Artemisia dracunculus, Boswellia sacra, Cinnamomum camphora, Citrus bergamia, Commiphora myrrha, Cymbopogon winterianus, Cymbopogon nardus, Dalbergia maritima, Helichrysum italicum, Jasminum polyanthum, Lavandula angustifolia, Matricaria chamomilla, Melaleuca alternifolia, Myrtus communis, Rosa damascena, Salvia sclarea, Santalum album, Verbena officinalis.*

**Table 2 molecules-28-01676-t002:** Essential oil composition (%) of the three *V. odorata* samples used in this study.

Compounds	RRI	% Similarity	MS and/or STD ^1^	*V. odorata* 1	*V. odorata* 2	*V. odorata* 3
α-Pinene	1016	96	MS, STD	-	-	0.9
Sabinene	1117	95	MS, STD	-	-	trace
Myrcene	1159	90	MS, STD	0.1	trace	trace
Limonene	1194	97	MS, STD	0.3	trace	0.4
Z-β-Ocimene	1232	86	MS, STD	0.1	0.1	-
Terpinolene	1281	95	MS, STD	0.1	trace	-
4-tert-Butylcyclohexyl acetate	1385	68	MS	-	0.4	-
Longifolene	1428	78	MS	-	-	0.4
Fenchyl acetate	1445	85	MS	-	-	0.6
Linalool	1541	89	MS, STD	0.2	0.6	-
Methyl-2-heptynoate	1543	90	MS	0.5	trace	-
*Lilia * ^2^	1548	75	MS	-	0.3	1.5
Terpinen-4-ol	1602	91	MS, STD	-	-	0.4
*p*-Menth-8-en-1-ol	1615	83	MS	-	-	3.9
Methyl benzoate	1624	86	MS	-	trace	-
cis-β-Terpineol	1630	84	MS	-	-	1.5
δ-Terpineol	1675	81	MS	-	-	0.3
α-Terpineol	1701	94	MS, STD	-	-	**38.51** ^3^
λ-Terpineol	1714	85	MS	-	-	9.1
Benzyl acetate	1734	89	MS, STD	1.0	1.0	**17.6**
α-Methyl benzyl acetate	1739	78	MS	0.3	0.7	-
*Methyl octane carbonate*	1745	79	MS	1.8	0.1	-
Nerol	1800	82	MS, STD	-	0.1	-
Phenethyl acetate	1807	87	MS	6.9	0.2	-
β-Ionone	1841	80	MS	5.4	0.4	1.4
Methyl ionone	1845	78	MS	6.7	0.5	0.8
α-Cetone	1851	86	MS			**13.8**
α-N-methyl ionone	1903	71	MS	-	-	2.8
α-Dimethyl phenyl acetate	1923	81	MS	-	0.2	-
Phenyl ethyl alcohol	1986	95	MS	3.4	1.5	-
α-N-methyl ionone	1903	74	MS	-	-	2.8
Phenethyl alcohol	1991	94	MS	**11.8**	**39.0**	1.2
*Triacetin*	2127	78	MS	-	0.8	-
*Isopropyl myristate*	2215	89	MS, STD	-	**42.7**	-
2-Nonynoic acid, methyl ester	2212	76	MS	**54.3**	-	-
**Total**				**92.9**	**87.9**	**95.1**

^1^ MS: Mass spectra; STD: standard; ^2^ Synthetic compounds italicized; ^3^ major compounds (>10%) given in bold.

**Table 3 molecules-28-01676-t003:** Antimicrobial activity (mg/mL) of essential oils against the selected pathogen reference strains (*n* = 3).

Essential Oil (Common Name)	Gram-Positive					Gram-Negative				Yeast
	Ca ^1^	Ef	Sa	Se	Sp	Ab	Ec	Pa	Kp	Cal
*Viola odorata* L. 1	**0.50 (±0.00)** ^2^	2.00 (±0.00)	2.67 (±1.15)	2.00 (±0.00)	4.00 (±0.00)	2.00 (±0.00)	4.00 (±0.00)	1.33 (±0.58)	4.00 (±0.00)	**0.67 (±0.29)**
*Viola odorata* L. 2	**0.67 (±0.29)**	2.00 (±0.00)	2.67 (±1.15)	2.67 (±1.15)	2.50 (±1.00)	3.33 (±1.15)	4.00 (±0.00)	1.50 (±0.58)	4.00 (±0.00)	**0.67 (±0.29)**
*Viola odorata* L. 3	2.00 (±0.00)	3.33 (±1.15)	2.00 (±0.00)	**1.00 (±0.00)**	2.40 (±0.89)	2.00 (±0.00)	2.67 (±1.15)	2.25 (±1.15)	6.67 (±2.31)	2.00 (±0.00)
*Achillea millefolium* Ledeb. (yarrow)	1.33 (±0.58)	3.00 (±1.15)	4.00 (±0.00)	2.67 (±1.15)	3.20 (±1.10)	2.00 (±0.00)	2.00 (±0.00)	1.20 (±0.50)	4.00 (±0.00)	2.00 (±0.00)
*Cinnamomum camphora* (L.) T.Nees & C.H.Eberm. (camphor)	1.33 (±0.58)	3.00 (±1.15)	5.33 (±2.31)	2.67 (±1.15)	4.00 (±2.45)	2.00 (±0.00)	3.33 (±1.15)	1.40 (±0.58)	4.00 (±0.00)	2.00 (±0.00)
*Citrus bergamia* Risso (bergamot)	**0.75 (±0.35)**	2.00 (±0.00)	2.67 (±1.15)	4.00 (±0.00)	3.20 (±1.10)	3.00 (±1.15)	4.00 (±0.00)	1.60 (±0.58)	5.33 (±2.31)	1.33 (±0.58)
*Commiphora myrrha* Engl. (myrrh)	**0.13 (±0.00)**	**1.00 (±0.00)**	3.33 (±1.15)	2.00 (±0.00)	1.80 (±0.45)	2.00 (±0.00)	2.00 (±0.00)	1.20 (±0.50)	2.67 (±1.15)	2.67 (±1.15)
*Cymbopogon citratus* Stapf (lemongrass)	**0.33 (±0.14)**	1.50 (±0.58)	2.00 (±0.00)	**0.67 (±0.29)**	1.40 (±0.55)	2.00 (±0.00)	2.00 (±0.00)	**0.60 (±0.25)**	3.33 (±1.15)	8.00 (±0.00)
*Cymbopogon nardus* (L.) Rendle (citronella)	**0.83 (±0.29)**	3.00 (±1.15)	2.67 (±1.15)	2.00 (±0.00)	2.20 (±1.10)	2.00 (±0.00)	2.67 (±1.15)	1.20 (±0.50)	6.67 (±2.31)	1.67 (±0.58)
*Eucalyptus globulus* Labill. (eucalyptus)	**0.10 (±0.04)**	2.00 (±0.00)	4.00 (±0.00)	2.00 (±0.00)	1.80 (±0.45)	3.00 (±1.15)	2.67 (±1.15)	1.60 (±0.58)	2.67 (±1.15)	**1.00 (±0.00)**
*Laurus nobilis* L. (bay)	1.33 (±0.58)	3.00 (±1.15)	4.00 (±0.00)	2.67 (±1.15)	2.00 (±0.00)	2.00 (±0.00)	3.33 (±1.15)	1.10 (±0.63)	4.00 (±0.00)	2.00 (±0.00)
*Lavandula angustifolia* Bubani (lavender)	1.33 (±0.58)	**1.00 (±0.00)**	4.00 (±0.00)	2.67 (±1.15)	1.80 (±0.45)	2.00 (±0.00)	2.00 (±0.00)	1.80 (±1.41)	4.00 (±0.00)	1.33 (±0.58)
*Leptospermum scoparium* J.R.Forst. and G.Forst (manuka)	**0.33 (±0.14)**	1.50 (±0.58)	2.00 (±0.00)	**0.67 (±0.29)**	1.20 (±0.45)	2.00 (±0.00)	1.33 (±0.58)	**0.80 (±0.25)**	3.33 (±1.15)	8.00 (±0.00)
*Litsea cubeba* Pers (may chang)	**0.21 (±0.07)**	1.50 (±0.58)	2.00 (±0.00)	**0.67 (±0.29)**	1.40 (±0.55)	2.00 (±0.00)	1.33 (±0.58)	**0.80 (±0.25)**	3.33 (±1.15)	8.00 (±0.00)
*Matricaria recutita* L. (German chamomile)	**0.75 (±0.35)**	1.50 (±0.58)	4.00 (±0.00)	1.33 (±0.58)	1.60 (±0.55)	2.00 (±0.00)	1.67 (±0.58)	**0.60 (±0.25)**	4.00 (±0.00)	2.67 (±1.15)
*Melaleuca alternifolia* Cheel (tea tree)	**0.83 (±0.29)**	2.00 (±0.00)	2.00 (±0.00)	2.67 (±1.15)	2.80 (±1.10)	2.00 (±0.00)	4.00 (±0.00)	1.20 (±0.50)	4.00 (±0.00)	1.33 (±0.58)
*Pogostemon patchouli* Benth. (patchouli)	**0.25 (±0.00)**	1.25 (±0.87)	2.67 (±1.15)	**0.50 (±0.00)**	1.40 (±0.55)	**1.00 (±0.00)**	2.00 (±0.00)	1.10 (±0.50)	4.00 (±0.00)	1.33 (±0.58)
*Rosmarinus officinalis* L. (rosemary)	**1.00 (±0.00)**	2.00 (±0.00)	4.00 (±0.00)	2.00 (±0.00)	2.00 (±0.00)	2.00 (±0.00)	3.00 (±1.73)	1.20 (±0.50)	2.67 (±1.15)	**1.00 (±0.00)**
*Salvia officinalis* Spreng. (sage)	**1.00 (±0.00)**	2.00 (±0.00)	3.33 (±1.15)	2.67 (±1.15)	2.40 (±0.89)	2.00 (±0.00)	3.33 (±1.15)	1.60 (±0.58)	2.67 (±1.15)	**1.00 (±0.00)**
*Santalum album* L. (sandalwood)	**0.09 (±0.04)**	**0.13 (±0.00)**	**0.50 (±0.00)**	**0.25 (±0.00)**	**0.70 (±0.27)**	**0.38 (±0.14)**	**0.50 (±0.00)**	**0.85 (±0.38)**	2.00 (±0.00)	1.33 (±0.58)
*S. austrocaledonicum* Vieill. (sandalwood)	**0.10 (±0.04)**	**0.50 (±0.00)**	**0.83 (±0.29)**	**0.33 (±0.14)**	**0.45 (±0.11)**	**0.50 (±0.00)**	**0.67 (±0.29)**	**1.00 (±0.00)**	2.00 (±0.00)	5.33 (±2.31)
*Vetiveria zizanioides* Stapf (vetiver)	**0.10 (±0.04)**	**0.75 (±0.29)**	1.33 (±0.58)	**0.67 (±0.29)**	**0.80 (±0.27)**	**0.75 (±0.29)**	1.33 (±0.58)	**0.80 (±0.25)**	2.67 (±1.15)	5.33 (±2.31)
*Zingiber officinale* Roscoe (ginger)	**0.58 (±0.38)**	4.00 (±0.00)	5.33 (±2.31)	2.67 (±1.15)	2.60 (±1.34)	2.00 (±0.00)	4.00 (±0.00)	1.40 (±0.58)	4.00 (±0.00)	2.00 (±0.00)
**Positive control (ciprofloxacin/amphotericin)**	6.25 × 10^−4^	1.56 × 10^−4^	3.125 × 10^−3^	3.125 × 10^−4^	1.56 × 10^−4^	1.56 × 10^−4^	3.9 × 10^−5^	3.9 × 10^−5^	3.125 × 10^−4^	1.56 × 10^−4^
**Negative control (water in acetone)**	>8.00	4.00	>8.00	>8.00	>8.00	>8.00	>8.00	>8.00	>8.00	>8.00
**Culture control**	>8.00	>8.00	>8.00	>8.00	>8.00	>8.00	>8.00	>8.00	>8.00	>8.00

^1^ Ca—*C. acnes* ATCC 6919; Ef—*E. faecium* ATCC 8739; Sa—*S. aureus* ATCC 25924; Se—*S. epidermis* ATCC 14990; Sp—*S. pyogenes* ATCC 12344; Ab—*A. baumannii* ATCC 17606; Ec—*E. coli* ATCC 25922; Kp—*K. pneumoniae* ATCC 13883; Pa—*P. aeruginosa* ATCC 27858; Cal—*C. albicans* ATCC 10231; ^2^ MIC value (mg/mL) (Standard deviation of three repeats given in brackets); noteworthy activity given in bold.

**Table 4 molecules-28-01676-t004:** Antimicrobial activity (mg/mL) of essential oils in combination with the different *V. odorata* chemotypes and ΣFIC values (*n* = 3).

**Essential Oil** **Combination**		**Gram-Positive**										**Gram-Negative**								**Yeast**	
		**Ca ^1^**		**Ef**		**Sa**		**Se**		**Sp**		**Ab**		**Ec**		**Pa**		**Kp**		**Cal**	
**Commercial EO**	** *V. odorata* ** **EO**	**MIC ^2^**	**ΣFIC ^3^**	**MIC**	**ΣFIC**	**MIC**	**ΣFIC**	**MIC**	**ΣFIC**	**MIC**	**ΣFIC**	**MIC**	**ΣFIC**	**MIC**	**ΣFIC**	**MIC**	**ΣFIC**	**MIC**	**ΣFIC**	**MIC**	**ΣFIC**
*A. millefolium*	1	**1.00 (±0.00)**	1.38	**1.00 (±0.00)**	** *0.42* **	2.00 (±0.00)	0.63	2.00 (±0.00)	0.88	4.00 (±0.00)	1.13	4.00 (±0.00)	2.00	8.00 (±0.00)	3.00	2.00 (±0.00)	1.58	4.00 (±0.00)	1.00	**1.00 (±0.00)**	1.00
	2	**1.00 (±0.00)**	1.12	4.00 (±0.00)	1.67	4.00 (±0.00)	1.25	4.00 (±0.00)	1.50	2.00 (±0.00)	0.71	2.00 (±0.00)	0.80	4.00 (±0.00)	1.50	2.00 (±0.00)	1.50	4.00 (±0.00)	1.00	3.00 (±1.41)	3.00
	3	1.50 (±0.71)	0.94	4.00 (±0.00)	1.27	8.00 (±0.00)	3.00	4.00 (±0.00)	2.75	2.00 (±0.00)	0.73	2.00 (±0.00)	1.00	2.00 (±0.00)	0.88	2.00 (±0.00)	1.28	4.00 (±0.00)	0.80	2.00 (±0.00)	1.00
*C. bergamia*	1	**0.75 (±0.35)**	1.25	4.00 (±0.00)	2.00	4.00 (±0.00)	1.50	2.00 (±0.00)	0.75	2.00 (±0.00)	0.56	2.00 (±0.00)	0.83	4.00 (±0.00)	1.00	2.00 (±0.00)	1.38	2.00 (±0.00)	** *0.44* **	**1.00 (±0.00)**	1.13
	2	1.50 (±0.71)	2.12	4.00 (±0.00)	2.00	4.00 (±0.00)	1.50	4.00 (±0.00)	1.25	2.00 (±0.00)	0.71	2.00 (±0.00)	0.63	8.00 (±0.00)	2.00	2.00 (±0.00)	1.29	2.00 (±0.00)	** *0.44* **	2.00 (±0.00)	2.25
	3	2.00 (±0.00)	1.83	4.00 (±0.00)	1.60	2.00 (±0.00)	0.88	4.00 (±0.00)	2.50	2.00 (±0.00)	0.73	2.00 (±0.00)	0.83	4.00 (±0.00)	1.25	**1.00 (±0.00)**	0.53	2.00 (±0.00)	** *0.34* **	4.00 (±0.00)	2.50
*C. camphora*	1	**1.00 (±0.00)**	1.38	**1.00 (±0.00)**	** *0.42* **	2.00 (±0.00)	0.56	2.00 (±0.00)	0.88	2.00 (±0.00)	** *0.50* **	8.00 (±0.00)	4.00	8.00 (±0.00)	2.20	2.00 (±0.00)	1.46	4.00 (±0.00)	1.00	2.00 (±0.00)	2.00
	2	**0.75 (±0.35)**	0.84	4.00 (±0.00)	1.67	4.00 (±0.00)	1.13	4.00 (±0.00)	1.50	2.00 (±0.00)	0.65	2.00 (±0.00)	0.80	4.00 (±0.00)	1.10	2.00 (±0.00)	1.38	4.00 (±0.00)	1.00	3.00 (±1.41)	3.00
	3	**1.00 (±0.00)**	0.63	4.00 (±0.00)	1.27	8.00 (±0.00)	2.75	4.00 (±0.00)	2.75	2.00 (±0.00)	0.67	2.00 (±0.00)	1.00	2.00 (±0.00)	0.68	2.00 (±0.00)	1.16	4.00 (±0.00)	0.80	4.00 (±0.00)	2.00
*C. citratus*	1	**0.25 (±0.00)**	0.63	**1.00 (±0.00)**	0.58	4.00 (±0.00)	1.75	2.00 (±0.00)	2.00	2.00 (±0.00)	0.96	2.00 (±0.00)	1.00	2.00 (±0.00)	0.75	2.00 (±0.00)	2.42	2.00 (±0.00)	0.55	**1.00 (±0.00)**	0.81
	2	**0.38 (±0.18)**	0.85	2.00 (±0.00)	1.17	2.00 (±0.00)	0.88	**1.00 (±0.00)**	0.94	**1.00 (±0.00)**	0.56	**1.00 (±0.00)**	** *0.40* **	4.00 (±0.00)	1.50	**1.00 (±0.00)**	1.17	2.00 (±0.00)	0.55	2.00 (±0.00)	1.63
	3	**0.75 (±0.35)**	1.31	**1.00 (±0.00)**	** *0.48* **	2.00 (±0.00)	1.00	**1.00 (±0.00)**	1.25	2.00 (±0.00)	1.13	**1.00 (±0.00)**	** *0.50* **	2.00 (±0.00)	0.88	**0.50 (±0.00)**	0.53	2.00 (±0.00)	** *0.45* **	2.00 (±0.00)	0.63
*C. myrrha*	1	**0.16 (±0.13)**	0.80	2.00 (±0.00)	1.50	4.00 (±0.00)	1.35	2.00 (±0.00)	1.00	2.00 (±0.00)	0.81	2.00 (±0.00)	1.00	2.00 (±0.00)	0.75	2.00 (±0.00)	1.58	2.00 (±0.00)	0.63	**1.00 (±0.00)**	0.94
	2	**1.00 (±0.00)**	4.75	2.00 (±0.00)	1.50	4.00 (±0.00)	1.35	2.00 (±0.00)	0.88	2.00 (±0.00)	0.96	2.00 (±0.00)	0.80	4.00 (±0.00)	1.50	2.00 (±0.00)	1.50	2.00 (±0.00)	0.63	2.00 (±0.00)	1.88
	3	**0.50 (±0.00)**	2.13	2.00 (±0.00)	1.30	2.00 (±0.00)	0.80	2.00 (±0.00)	1.50	2.00 (±0.00)	0.97	**1.00 (±0.00)**	** *0.50* **	2.00 (±0.00)	0.88	**1.00 (±0.00)**	0.64	2.00 (±0.00)	0.53	2.00 (±0.00)	0.88
*C. nardus*	1	**1.00 (±0.00)**	1.60	**1.00 (±0.00)**	0.42	3.00 (±1.41)	1.13	2.00 (±0.00)	1.00	4.00 (±0.00)	1.41	4.00 (±0.00)	2.00	4.00 (±0.00)	1.25	4.00 (±0.00)	3.17	2.00 (±0.00)	** *0.40* **	2.00 (±0.00)	2.10
	2	**1.00 (±0.00)**	1.35	4.00 (±0.00)	1.67	4.00 (±0.00)	1.50	2.00 (±0.00)	0.88	2.00 (±0.00)	0.85	2.00 (±0.00)	0.80	4.00 (±0.00)	1.25	**1.00 (±0.00)**	0.75	2.00 (±0.00)	** *0.40* **	4.00 (±0.00)	4.20
	3	1.50 (±0.71)	1.28	2.00 (±0.00)	0.63	4.00 (±0.00)	1.75	2.00 (±0.00)	1.50	2.00 (±0.00)	0.87	2.00 (±0.00)	1.00	2.00 (±0.00)	0.75	**1.00 (±0.00)**	0.64	2.00 (±0.00)	** *0.30* **	2.00 (±0.00)	1.10
*E. globulus*	1	**0.19 (±0.09)**	1.10	2.00 (±0.00)	1.00	4.00 (±0.00)	1.25	4.00 (±0.00)	2.00	2.00 (±0.00)	0.81	2.00 (±0.00)	0.83	4.00 (±0.00)	1.25	4.00 (±0.00)	2.75	2.00 (±0.00)	0.63	**1.00 (±0.00)**	1.25
	2	**0.38 (±0.18)**	2.11	4.00 (±0.00)	2.00	4.00 (±0.00)	1.25	4.00 (±0.00)	1.75	2.00 (±0.00)	0.96	2.00 (±0.00)	0.63	4.00 (±0.00)	1.25	2.00 (±0.00)	1.29	2.00 (±0.00)	0.63	2.00 (±0.00)	2.50
	3	**0.38 (±0.18)**	1.92	4.00 (±0.00)	1.60	2.00 (±0.00)	0.75	4.00 (±0.00)	3.00	2.00 (±0.00)	0.97	2.00 (±0.00)	0.83	3.00 (±1.41)	1.13	**1.00 (±0.00)**	0.53	4.00 (±0.00)	1.05	2.00 (±0.00)	1.50
*L. angastifolia*	1	**1.00 (±0.00)**	1.38	2.00 (±0.00)	1.50	4.00 (±0.00)	1.25	2.00 (±0.00)	0.88	2.00 (±0.00)	0.81	2.00 (±0.00)	1.00	2.00 (±0.00)	0.75	2.00 (±0.00)	1.31	2.00 (±0.00)	** *0.50* **	**1.00 (±0.00)**	1.13
	2	**0.75 (±0.35)**	0.84	4.00 (±0.00)	3.00	4.00 (±0.00)	1.25	2.00 (±0.00)	0.75	2.00 (±0.00)	0.96	2.00 (±0.00)	0.80	4.00 (±0.00)	1.50	**1.00 (±0.00)**	0.61	2.00 (±0.00)	** *0.50* **	2.00 (±0.00)	2.25
	3	2.00 (±0.00)	1.25	4.00 (±0.00)	2.60	2.00 (±0.00)	0.75	2.00 (±0.00)	1.38	2.00 (±0.00)	0.97	2.00 (±0.00)	1.00	4.00 (±0.00)	1.75	**1.00 (±0.00)**	**0.50**	4.00 (±0.00)	0.80	2.00 (±0.00)	1.25
*L. cubeba*	1	**0.09 (±0.04)**	** *0.31* **	**1.00 (±0.00)**	0.58	2.00 (±0.00)	0.88	2.00 (±0.00)	2.00	2.00 (±0.00)	0.96	2.00 (±0.00)	1.00	2.00 (±0.00)	1.00	2.00 (±0.00)	2.00	2.00 (±0.00)	0.55	**1.00 (±0.00)**	0.81
	2	**0.75 (±0.35)**	2.36	4.00 (±0.00)	2.33	2.00 (±0.00)	0.88	**1.00 (±0.00)**	0.94	**1.00 (±0.00)**	0.56	2.00 (±0.00)	0.80	4.00 (±0.00)	2.00	**1.00 (±0.00)**	0.96	2.00 (±0.00)	0.55	2.00 (±0.00)	1.63
	3	1.50 (±0.71)	3.98	**1.00 (±0.00)**	** *0.48* **	2.00 (±0.00)	1.00	**1.00 (±0.00)**	1.25	2.00 (±0.00)	1.13	**1.00 (±0.00)**	0.50	2.00 (±0.00)	1.13	**0.50 (±0.00)**	** *0.42* **	2.00 (±0.00)	** *0.45* **	2.00 (±0.00)	0.63
** *L. nobilis* **	1	**1.00 (±0.00)**	1.38	2.00 (±0.00)	0.83	4.00 (±0.00)	1.25	4.00 (±0.00)	1.75	4.00 (±0.00)	1.50	2.00 (±0.00)	1.00	4.00 (±0.00)	1.10	2.00 (±0.00)	1.66	2.00 (±0.00)	** *0.50* **	2.00 (±0.00)	2.00
	2	**1.00 (±0.00)**	1.12	4.00 (±0.00)	1.67	4.00 (±0.00)	1.25	2.00 (±0.00)	0.75	2.00 (±0.00)	0.90	2.00 (±0.00)	0.80	4.00 (±0.00)	1.10	**1.00 (±0.00)**	0.79	4.00 (±0.00)	1.00	4.00 (±0.00)	4.00
	3	2.00 (±0.00)	1.25	2.00 (±0.00)	0.63	4.00 (±0.00)	1.50	2.00 (±0.00)	1.38	2.00 (±0.00)	0.92	2.00 (±0.00)	1.00	8.00 (±0.00)	2.70	2.00 (±0.00)	1.35	4.00 (±0.00)	0.80	2.00 (±0.00)	1.00
*L. scoparium*	1	**0.50 (±0.00)**	1.25	**1.00 (±0.00)**	0.58	2.00 (±0.00)	0.88	**1.00 (±0.00)**	1.00	2.00 (±0.00)	1.08	2.00 (±0.00)	1.00	4.00 (±0.00)	2.00	4.00 (±0.00)	4.00	2.00 (±0.00)	0.55	**1.00 (±0.00)**	0.81
	2	**1.00 (±0.00)**	2.25	4.00 (±0.00)	2.33	2.00 (±0.00)	0.88	**1.00 (±0.00)**	0.94	2.00 (±0.00)	1.23	2.00 (±0.00)	0.80	4.00 (±0.00)	2.00	**0.50 (±0.00)**	** *0.48* **	2.00 (±0.00)	0.55	2.00 (±0.00)	1.63
	3	**1.00 (±0.00)**	1.75	**1.00 (±0.00)**	** *0.48* **	4.00 (±0.00)	2.00	**1.00 (±0.00)**	1.25	2.00 (±0.00)	1.25	**1.00 (±0.00)**	** *0.50* **	2.00 (±0.00)	1.13	**0.50 (±0.00)**	** *0.42* **	2.00 (±0.00)	** *0.45* **	2.00 (±0.00)	0.63
*M. alternifolia*	1	**0.50 (±0.00)**	0.80	2.00 (±0.00)	1.00	4.00 (±0.00)	1.75	2.00 (±0.00)	0.88	4.00 (±0.00)	1.21	2.00 (±0.00)	1.00	4.00 (±0.00)	1.00	2.00 (±0.00)	1.58	2.00 (±0.00)	** *0.50* **	**1.00 (±0.00)**	1.13
	2	**0.75 (±0.35)**	1.01	2.00 (±0.00)	1.00	4.00 (±0.00)	1.75	4.00 (±0.00)	1.50	2.00 (±0.00)	0.76	4.00 (±0.00)	1.60	4.00 (±0.00)	1.00	2.00 (±0.00)	1.50	2.00 (±0.00)	** *0.50* **	2.00 (±0.00)	2.25
	3	**1.00 (±0.00)**	0.85	4.00 (±0.00)	1.60	**1.00 (±0.00)**	** *0.50* **	4.00 (±0.00)	2.75	2.00 (±0.00)	0.77	2.00 (±0.00)	1.00	2.00 (±0.00)	0.63	2.00 (±0.00)	1.28	2.00 (±0.00)	** *0.40* **	2.00 (±0.00)	1.25
*M. recutita*	1	**0.75 (±0.35)**	1.25	**1.00 (±0.00)**	0.58	4.00 (±0.00)	1.25	2.00 (±0.00)	1.25	2.00 (±0.00)	0.88	2.00 (±0.00)	1.00	2.00 (±0.00)	0.85	2.00 (±0.00)	2.42	2.00 (±0.00)	** *0.50* **	**1.00 (±0.00)**	0.94
	2	**0.38 (±0.18)**	0.54	4.00 (±0.00)	2.33	4.00 (±0.00)	1.25	2.00 (±0.00)	1.13	4.00 (±0.00)	2.05	2.00 (±0.00)	0.80	4.00 (±0.00)	1.70	1.50 (±0.71)	1.75	2.00 (±0.00)	** *0.50* **	4.00 (±0.00)	3.75
	3	**0.75 (±0.35)**	0.69	2.00 (±0.00)	0.97	2.00 (±0.00)	0.75	2.00 (±0.00)	1.75	2.00 (±0.00)	1.04	2.00 (±0.00)	1.00	2.00 (±0.00)	0.98	**1.00 (±0.00)**	1.06	2.00 (±0.00)	** *0.40* **	2.00 (±0.00)	0.88
*P. patchouli*	1	**1.00 (±0.00)**	3.00	8.00 (±0.00)	5.20	4.00 (±0.00)	1.50	2.00 (±0.00)	1.50	2.00 (±0.00)	0.96	2.00 (±0.00)	1.50	2.00 (±0.00)	0.75	2.00 (±0.00)	1.66	2.00 (±0.00)	** *0.50* **	**1.00 (±0.00)**	1.13
	2	**1.00 (±0.00)**	2.75	**1.00 (±0.00)**	0.65	4.00 (±0.00)	1.50	2.00 (±0.00)	1.38	2.00 (±0.00)	1.11	2.00 (±0.00)	1.30	4.00 (±0.00)	1.50	**1.00 (±0.00)**	0.79	4.00 (±0.00)	1.00	2.00 (±0.00)	2.25
	3	**0.50 (±0.00)**	1.13	2.00 (±0.00)	1.10	2.00 (±0.00)	0.88	2.00 (±0.00)	2.00	2.00 (±0.00)	1.13	**1.00 (±0.00)**	0.75	4.00 (±0.00)	1.75	**1.00 (±0.00)**	0.68	4.00 (±0.00)	0.80	2.00 (±0.00)	1.25
*R. officinalis*	1	**1.00 (±0.00)**	1.50	4.00 (±0.00)	2.00	4.00 (±0.00)	1.25	2.00 (±0.00)	1.00	2.00 (±0.00)	0.75	2.00 (±0.00)	1.00	4.00 (±0.00)	1.17	2.00 (±0.00)	1.58	2.00 (±0.00)	0.63	2.00 (±0.00)	2.50
	2	1.50 (±0.71)	1.87	4.00 (±0.00)	2.00	4.00 (±0.00)	1.25	2.00 (±0.00)	0.88	2.00 (±0.00)	0.90	2.00 (±0.00)	0.80	4.00 (±0.00)	1.17	1.00 (±0.00)	0.75	2.00 (±0.00)	0.63	2.00 (±0.00)	2.50
	3	2.00 (±0.00)	1.50	4.00 (±0.00)	1.60	2.00 (±0.00)	0.75	2.00 (±0.00)	1.50	2.00 (±0.00)	0.92	2.00 (±0.00)	1.00	4.00 (±0.00)	1.42	1.00 (±0.00)	0.64	4.00 (±0.00)	1.05	4.00 (±0.00)	3.00
*S. album*	1	**0.25 (±0.00)**	1.58	**0.25 (±0.00)**	1.06	2.00 (±0.00)	2.38	**1.00 (±0.00)**	2.25	**1.00 (±0.00)**	0.84	**0.50 (±0.00)**	0.79	2.00 (±0.00)	2.25	2.00 (±0.00)	1.93	2.00 (±0.00)	0.75	**0.50 (±0.00)**	0.56
	2	**0.25 (±0.00)**	1.52	**0.25 (±0.00)**	1.06	2.00 (±0.00)	2.38	2.00 (±0.00)	4.38	**0.25 (±0.00)**	** *0.23* **	**1.00 (±0.00)**	1.48	4.00 (±0.00)	4.50	**0.50 (±0.00)**	** *0.46* **	4.00 (±0.00)	1.50	4.00 (±0.00)	4.50
	3	**0.50 (±0.00)**	2.79	**0.50 (±0.00)**	2.08	**0.50 (±0.00)**	0.63	2.00 (±0.00)	5.00	**0.50 (±0.00)**	0.46	**0.25 (±0.00)**	** *0.40* **	**1.00 (±0.00)**	1.19	**0.50 (±0.00)**	** *0.41* **	2.00 (±0.00)	0.65	**0.50 (±0.00)**	** *0.31* **
*S. austrocaledonicum*	1	**0.25 (±0.00)**	1.45	**0.25 (±0.00)**	** *0.31* **	**1.00 (±0.00)**	0.79	4.00 (±0.00)	7.00	**0.50 (±0.00)**	0.62	**0.50 (±0.00)**	0.63	**1.00 (±0.00)**	0.88	**1.00 (±0.00)**	0.88	2.00 (±0.00)	0.75	**1.00 (±0.00)**	0.84
	2	**0.19 (±0.09)**	1.05	**0.25 (±0.00)**	** *0.31* **	1.50 (±0.71)	1.18	4.00 (±0.00)	6.75	**0.25 (±0.00)**	** *0.33* **	**0.50 (±0.00)**	0.58	**1.00 (±0.00)**	0.88	**0.50 (±0.00)**	** *0.42* **	4.00 (±0.00)	1.50	2.00 (±0.00)	1.69
	3	**0.13 (±0.00)**	0.66	**1.00 (±0.00)**	1.15	**0.50 (±0.00)**	** *0.43* **	4.00 (±0.00)	8.00	**0.50 (±0.00)**	0.66	**0.25 (±0.00)**	** *0.31* **	**1.00 (±0.00)**	0.94	**0.25 (±0.00)**	** *0.18* **	4.00 (±0.00)	1.30	**1.00 (±0.00)**	** *0.34* **
*S. officinalis*	1	**1.00 (±0.00)**	1.50	4.00 (±0.00)	2.00	4.00 (±0.00)	1.35	2.00 (±0.00)	0.88	4.00 (±0.00)	1.33	2.00 (±0.00)	1.00	2.00 (±0.00)	0.55	2.00 (±0.00)	1.38	2.00 (±0.00)	0.63	2.00 (±0.00)	2.50
	2	**1.00 (±0.00)**	1.25	4.00 (±0.00)	2.00	4.00 (±0.00)	1.35	4.00 (±0.00)	1.50	2.00 (±0.00)	0.82	2.00 (±0.00)	0.80	4.00 (±0.00)	1.10	2.00 (±0.00)	1.29	4.00 (±0.00)	1.25	2.00 (±0.00)	2.50
	3	2.00 (±0.00)	1.50	2.00 (±0.00)	0.80	2.00 (±0.00)	0.80	4.00 (±0.00)	2.75	2.00 (±0.00)	0.83	2.00 (±0.00)	1.00	4.00 (±0.00)	1.35	2.00 (±0.00)	1.07	2.00 (±0.00)	0.53	2.00 (±0.00)	1.50
*V. zizanioides*	1	**0.38 (±0.18)**	2.20	8.00 (±0.00)	7.33	2.00 (±0.00)	1.13	**1.00 (±0.00)**	1.00	**1.00 (±0.00)**	0.75	2.00 (±0.00)	1.83	2.00 (±0.00)	1.00	2.00 (±0.00)	2.00	2.00 (±0.00)	0.63	**1.00 (±0.00)**	0.84
	2	**0.75 (±0.35)**	4.16	**1.00 (±0.00)**	0.92	4.00 (±0.00)	2.25	2.00 (±0.00)	1.88	**1.00 (±0.00)**	0.83	2.00 (±0.00)	1.63	4.00 (±0.00)	2.00	**0.50 (±0.00)**	** *0.48* **	4.00 (±0.00)	1.25	2.00 (±0.00)	1.69
	3	**0.13 (±0.00)**	0.66	2.00 (±0.00)	1.63	4.00 (±0.00)	2.50	2.00 (±0.00)	2.50	**1.00 (±0.00)**	0.83	**1.00 (±0.00)**	0.92	4.00 (±0.00)	2.25	**0.50 (±0.00)**	** *0.42* **	4.00 (±0.00)	1.05	2.00 (±0.00)	0.69
*Z. officinale*	1	**0.50 (±0.00)**	0.93	4.00 (±0.00)	1.50	3.00 (±1.41)	0.84	4.00 (±0.00)	1.75	4.00 (±0.00)	1.27	2.00 (±0.00)	1.00	4.00 (±0.00)	1.00	4.00 (±0.00)	2.93	4.00 (±0.00)	1.00	**0.50 (±0.00)**	** *0.50* **
	2	**1.00 (±0.00)**	1.60	4.00 (±0.00)	1.50	4.00 (±0.00)	1.13	2.00 (±0.00)	0.75	2.00 (±0.00)	0.78	2.00 (±0.00)	0.80	4.00 (±0.00)	1.00	**1.00 (±0.00)**	0.69	2.00 (±0.00)	** *0.50* **	2.00 (±0.00)	2.00
	3	**1.00 (±0.00)**	1.11	2.00 (±0.00)	0.55	4.00 (±0.00)	1.38	2.00 (±0.00)	1.38	2.00 (±0.00)	0.80	2.00 (±0.00)	1.00	2.00 (±0.00)	0.63	**1.00 (±0.00)**	0.58	4.00 (±0.00)	0.80	2.00 (±0.00)	1.00

^1^ Ca—*C. acnes* ATCC 6919; Ef—*E. faecium* ATCC 8739; Sa—*S. aureus* ATCC 25924; Se—*S. epidermis* ATCC 14990; Sp—*S. pyogenes* ATCC 12344; Ab—*A. baumannii* ATCC 17606; Ec—*E. coli* ATCC 25922; Kp—*K. pneumoniae* ATCC 13883; Pa—*P. aeruginosa* ATCC 27858; Cal—*C. albicans* ATCC 10231. ^2^ Noteworthy activities given in bold; ^3^ Synergy reported in bold italics.

## Data Availability

Not applicable.

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
