# Peer review of "Commercially Available Viola odorata Oil, Chemical Variability and Antimicrobial Activity"

_molecules, 2023, doi:10.3390/molecules28041676_

Round 1
Reviewer 1 Report
The paper is clearly written and methods are well described. Nevertheless, presentation of results should be improved. Tables are too rich in data and difficult to read, and figure 1 should be better described.
Author Response
The paper is clearly written and methods are well described. Nevertheless, presentation of results should be improved. Tables are too rich in data and difficult to read, and figure 1 should be better described.
We agree with the reviewer that the tables are rich in data. Unfortunately, any amendments would mean removing pertinent data that would misconstrue the final take home message. Figure 1 acts as a summary to simplify this data and hence has been better explained for clarity (page 6)
Reviewer 2 Report
The MS entitled “Commercially available Viola odorata oil, chemical variability and antimicrobial activity” was reviewed. The authors have summarized their research work on the determination of antimicrobial activity of three Viola odorata samples, in combination with 20 different commercial essential oils. The work seems to be noteworthy and the data seems to be original. However, certain queries should be addressed before any decision should be made.
1. Why the authors were prompted to study combination of EOs?
2. Are the targeted bacteria resistant?
3. Is there any aromatherapy currently in practice as antibiotic against pathogenic bacteria? If yes, then kindly update the literature. Also state any combine essential oils used as antibiotics.
4. Why the authors did not perform any isolation of Viola essential oils themselves? Were the plants not available in their regions? It should be better if instead using some available market essential oils, the pure isolates should be used for combine analysis and activities. Why authors failed in producing their own oils?
5. Only MIC is not sufficient to make any therapeutic guess. The IC50 comparative studies are recommended to find potentially new drug. How the authors respond to that?
6. is there any medicinal related commercial/ industrial purpose the study aims to achieve?
Author Response
The MS entitled “Commercially available Viola odorata oil, chemical variability and antimicrobial activity” was reviewed. The authors have summarized their research work on the determination of antimicrobial activity of three Viola odorata samples, in combination with 20 different commercial essential oils. The work seems to be noteworthy and the data seems to be original. However, certain queries should be addressed before any decision should be made.
- Why the authors were prompted to study combination of EOs?
Essential oils are frequently used in combination with the goal of achieving therapeutic synergy. While Viola is often recommended in a combination (Table 1), little validation has been done to scientifically validate the improved efficacy in combination. Text has been added to the introduction (page 2) to clarify this.
- Are the targeted bacteria resistant?
The micro-organisms are the ones involved in the diseases for which V. odorata is often recommended. The strains used are reference strains. This has been clarified on page 9.
- Is there any aromatherapy currently in practice as antibiotic against pathogenic bacteria? If yes, then kindly update the literature. Also state any combine essential oils used as antibiotics.
Indeed this aspect has been well studied. This has been addressed in the introduction with reference to scientific articles. References 6, 13, 29 and 45 represent some of our earlier studies on the subject matter and have been incorporated within the manuscript where applicable.
- Why the authors did not perform any isolation of Viola essential oils themselves? Were the plants not available in their regions? It should be better if instead using some available market essential oils, the pure isolates should be used for combine analysis and activities. Why authors failed in producing their own oils?
Viola is unfortunately not readily grown in South Africa. Furthermore, the commercially available product is recommended within the aromatherapeutic literature. Hence, the purchase of samples for further study was warranted.
- Only MIC is not sufficient to make any therapeutic guess. The IC50 comparative studies are recommended to find potentially new drug. How the authors respond to that?
From an antimicrobial perspective the MIC is a scientifically accepted means of measuring antimicrobial activity. I quote directly from Kalemba and Kunicka, “We strongly suggest publishing only the results of research where MIC/MBC values have been established” Kalemba D., Kunicka A. Antibacterial and antifungal properties of essential oils. Current Medicinal Chemistry 10, 813-829, 2003. One needs to keep in mind that the MIC considers inhibition of 100% of micro-organisms, as opposed to just 50%. Inhibition of 50% of micro-organisms means survival for the other 50%, thus leading to pathogenicity, mutation and resistance.
- is there any medicinal related commercial/ industrial purpose the study aims to achieve?
A therapeutic potential is identified. Historically, essential oils have been used readily for treating infections, however, natural products have been set aside as priority since the emergence of antibiotics. With the scourge of antimicrobial resistance now causing a worldwide problem and the lack of new antimicrobials becoming available, research is now including the previously neglected natural products. To rationalise the selection, we use the literature of aromatherapists/ those that still make frequent use of essential oils alone and in combination to identify adequate starting points for identifying entities that could validate use.
Various text in the introduction has been amended to articulate this more clearly.
Reviewer 3 Report
Abstract
The first idea of the abstract needs to be clarified. The authors say, “Viola odorata L. is frequently cited in the aromatherapeutic literature,” but what is cited? The specie, the essential oil, or what? I suppose the background of the title refers to essential oil, but it is crucial to clarify the idea. Furthermore, don't demerit your work by saying, “evidence validating the antimicrobial activity of this oil is lacking” change the idea for the antimicrobial activity; it is not fully explored or something like that.
When the authors say “three Viola odorata samples” again, you need to clarify whether that, Is a plant tissue? Oil? Essential oil?
I recommend that the authors rewrite the abstract. First, explain the most relevant of their work, in this case, the antimicrobial activity against the ESKAPE set, then explain the reason for this activity and the correlation ship with the chemical profile, finally talk about the discrepancies of the results in the literature and finally, your conclusions. Please, change the abstract with this idea.
Introduction
The first sentence in the introduction is good, but in the second sentence, when the authors say, “holistic way of healing the mind, body, and soul,” please, I really recommend writing in a scientific context. Aromatherapy with essential oils has a scientific background in different ways than the soul. Please and strongly recommend changing this sentence for different applications of essential oil in the science aspect. I am entirely sure that you will find enough literature to rewrite this idea.
The authors mentioned that the scientific evidence validating the antimicrobial activity is deficient. In this section, my question is Who scientific evidence? Please provide more references about this idea, and clarify What is deficient? Compared with what? And the authors need to propose some concentration.
Table 1 is attractive. Now that I understand the idea that the authors intend to explain in the abstract, please, add part of this idea in the abstract, maybe, the best combination.
Results and discussion
In the chemical analysis section, the authors must explain how to detect the analytes in Table 2. For example, if only compared with NIST Database, add in table 2 the % of coincidence. Why didn't you use an internal standard or alkane to establish the Kovax index? One of the compounds in table 2 take my attention, isopropyl myristate; this compound is used in the industry as a solvent for essential oil. Please justify why you add this compound to the table. Finally, why the sum of % is not 100%? Please clarify.
The authors say, "This is not the first study to identify fatty acids in Viola spp" in your case, you don't identify a fatty acid. Isopropyl myristate is a fatty acid derivative; it is, in this part, the root of my question because you need to derivate the meristic acid. Please review the literature and the biosynthesis of fatty acids to understand the complexity of saying that Isopropyl myristate is a natural product or justify with other studies that detected the same compound and the explication.
The antimicrobial activity section is good and has exciting discussions.
The conclusion is reasonable and according to the aim of the work.
According to the comments, I suggest that the author attend to the comments and resubmit for another review round. For that, I suggest Major Revision.
Author Response
The first idea of the abstract needs to be clarified. The authors say, “Viola odorata L. is frequently cited in the aromatherapeutic literature,” but what is cited? The specie, the essential oil, or what? I suppose the background of the title refers to essential oil, but it is crucial to clarify the idea. Furthermore, don't demerit your work by saying, “evidence validating the antimicrobial activity of this oil is lacking” change the idea for the antimicrobial activity; it is not fully explored or something like that.
The authors appreciate the affirmation and have now amended the abstract accordingly. Please see amended text as follows:
“Viola odorata L. oil is frequently recommended in the aromatherapeutic literature for the use in treating respiratory, urinary, and skin infections, however, antimicrobial evidence is lacking. Added to this, in aromatherapy, combinations of essential oils are predominantly utilized with the goal of achieving therapeutic synergy, yet no studies investigating the interaction of essential oil combinations with V. odorata oil exists. This study thus aimed to address these gaps by investigating the antimicrobial activity of three Viola odorata oil samples, sourced from different suppliers, independently and in combination with 20 different commercial essential oils.”
When the authors say “three Viola odorata samples” again, you need to clarify whether that, Is a plant tissue? Oil? Essential oil?
The authors have avoided using the word essential oils, as the chemical characterization revealed a number of added synthetic compounds. To clarify the Viola odorata samples are simply referred to as “oils” throughout the manuscript.
I recommend that the authors rewrite the abstract. First, explain the most relevant of their work, in this case, the antimicrobial activity against the ESKAPE set, then explain the reason for this activity and the correlation ship with the chemical profile, finally talk about the discrepancies of the results in the literature and finally, your conclusions. Please, change the abstract with this idea.
The abstract predominantly flows in this order, however, the ESKAPE pathogen set has not been addressed. As the study focused on a number of pathogens other than that of the ESKAPE group, we felt that to focus on this would be misrepresented in the body of the text.
Introduction
The first sentence in the introduction is good, but in the second sentence, when the authors say, “holistic way of healing the mind, body, and soul,” please, I really recommend writing in a scientific context. Aromatherapy with essential oils has a scientific background in different ways than the soul. Please and strongly recommend changing this sentence for different applications of essential oil in the science aspect. I am entirely sure that you will find enough literature to rewrite this idea.
Thank you for the recommendation, the introduction has been updated accordingly.
The authors mentioned that the scientific evidence validating the antimicrobial activity is deficient. In this section, my question is Who scientific evidence? Please provide more references about this idea, and clarify What is deficient? Compared with what? And the authors need to propose some concentration.
This has been elaborated and clarified in the introduction.
The proposed concentration is determined via the microdilution assay and a cut-off for noteworthy is defined as 1.00 mg/ml in the methods 3.3.2.
Table 1 is attractive. Now that I understand the idea that the authors intend to explain in the abstract, please, add part of this idea in the abstract, maybe, the best combination.
The abstract has been modified accordingly
Results and discussion
In the chemical analysis section, the authors must explain how to detect the analytes in Table 2. For example, if only compared with NIST Database, add in table 2 the % of coincidence. Why didn't you use an internal standard or alkane to establish the Kovax index? One of the compounds in table 2 take my attention, isopropyl myristate; this compound is used in the industry as a solvent for essential oil. Please justify why you add this compound to the table. Finally, why the sum of % is not 100%? Please clarify.
The compounds were identified by comparing the mass spectra with NIST®, Mass Finder® libraries. In addition, some authentic standards were also used as indicated Table 2. The sum is not 100% because some peaks (unknown compounds) could not be identified.
The percentage similarity for each compounds was added in the Table 2. The RRI calculated using the alkane mixture was also added.
The authors say, "This is not the first study to identify fatty acids in Viola spp" in your case, you don't identify a fatty acid. Isopropyl myristate is a fatty acid derivative; it is, in this part, the root of my question because you need to derivate the meristic acid. Please review the literature and the biosynthesis of fatty acids to understand the complexity of saying that Isopropyl myristate is a natural product or justify with other studies that detected the same compound and the explication.
Thank you for this valid comment. Artificial compounds have now been identified within Table 2. We have reviewed the compounds and italicised all non-essential oil compounds. We have also confirmed the presence of Isopropyl myristate by injecting a standard.
The antimicrobial activity section is good and has exciting discussions.
The conclusion is reasonable and according to the aim of the work.
Round 2
Reviewer 3 Report
The authors attended to all my comments and responded appropriately to each of them.
I have a few recommendations:
In the abstract, when the authors say "antimicrobial activity," I recommend highlighting the group of microbials. Just a few words about the complexity of this microbial and the importance of searching for new treatments or natural drugs.
The authors improve the introduction, and now the aim and research question are compressible.
Below table 2, the authors indicate the instructions to understand the table; however, they don't use the numbers. Please include the number in each part that corresponds.
Please, try to improve the resolution of figure 1.
The authors present an excellent work that meets the scope and goals of the journal. For that, I recommend publishing after minor revisions.
Author Response
The authors attended to all my comments and responded appropriately to each of them.
I have a few recommendations:
- In the abstract, when the authors say "antimicrobial activity," I recommend highlighting the group of microbials. Just a few words about the complexity of this microbial and the importance of searching for new treatments or natural drugs.
The section has been amended and now reads (new part highlighted in yellow font in manuscript) “This study thus aimed to address these gaps by investigating the antimicrobial activity of three Viola odorata oil samples, sourced from different suppliers, independently and in combination with 20 different commercial essential oils, against micro-organisms involved in respiratory, skin, and urinary tract infections associated with global resistance trends. These pathogens make up the ESKAPE group (Enterococcus faecium, Staphylococcus aureus, Klebsiella pneumoniae, Acinetobacter baumannii, Pseudomonas aeruginosa and Enterobacter spp.).
- Below table 2, the authors indicate the instructions to understand the table; however, they don't use the numbers. Please include the number in each part that corresponds.
The numbers have been included and are highlighted in the manuscript to allow for easier view.
- Please, try to improve the resolution of figure 1.
Separate high resolution figures have been included to replace figures during final editing process